# Osteoporosis induced by cellular senescence: A mathematical model

**Nourridine Siewe** [1]*, **Avner Friedman** [2]

**1** School of Mathematics and Statistics, College of Science, Rochester Institute of Technology, Rochester, New York, United States of America, **2** Department of Mathematics, The Ohio State University, Columbus, Ohio, United States of America

* nxssma@rit.edu

## Abstract

Osteoporosis is a disease characterized by loss of bone mass, where bones become fragile and more likely to fracture. Bone density begins to decrease at age 50, and a state of osteoporosis is defined by loss of more than 25%. Cellular senescence is a permanent arrest of normal cell cycle, while maintaining cell viability. The number of senescent cells increase with age. Since osteoporosis is an aging disease, it is natural to consider the question to what extend senescent cells induce bone density loss and osteoporosis. In this paper we use a mathematical model to address this question. We determine the percent of bone loss for men and women during age 50 to 100 years, and the results depend on the rate $\eta$ of net formation of senescent cell, with $\eta = 1$ being the average rate. In the case $\eta = 1$, the model simulations are in agreement with empirical data. We also consider senolytic drugs, like fisetin and quercetin, that selectively eliminate senescent cells, and assess their efficacy in terms of reducing bone loss. For example, at $\eta = 1$, with estrogen hormonal therapy and early treatment with fisetin, bone density loss for women by age 75 is 23.4% (below osteoporosis), while with no treatment with fisetin it is 25.8% (osteoporosis); without even a treatment with estrogen hormonal therapy, bone loss of 25.3% occurs already at age 65.

## 1 Introduction

Osteoporosis is a disease characterized by bone mass loss, where bones become fragile and more likely to fracture. A state of osteoporosis is defined by a loss of more than 25% of average bone density at age 30 [1]. The risk of developing osteoporosis increases as people grow older. Bone density is almost unchanged between 30 and 50 years of age, and then it begins to decrease [2] Fig 6.23. Based on [2] Fig 6.23, we assume that by age 80, bone density has decreased by 13–15% for men and 35–40% for women. In a 2015–2018 study [3] it was found that the probability of fracture is 1% at ages 50–65 years, and it increases to 1.5% at 70, 3.5% at 80, 5.5% at 90, and 8.5% at age 95; 1.5 times more in women than in men. Another 2022 study [4] estimates that one-tenth of women aged 60, one-fifth aged 70, two-fifths aged 80 and two-thirds aged 90 carry the diagnosis of osteoporosis.

Although there are several drugs for the treatment of osteoporosis, it may be desirable to treat vulnerable people by drugs that can delay the onset of osteoporosis. In the present paper we assess the efficacy of such drugs.

**Data Availability Statement:** All relevant data are within the manuscript.

**Funding:** The author(s) received no specific funding for this work.

**Competing interests:** The authors have declared that no competing interests exist.

Cellular senescence is a permanent arrest of normal cell cycle, while maintaining cell viability. The number of senescent cells increases with age. Senescence of stem cells and progenitor cells drive tissue aging. In particular, senescence in aging bone leads to osteoporosis [5, 6].

Senescence in stem cells compromises their ability to differentiate [7, 8]. Senescence stem cells secrete senescence-associated secretory phenotype (SASP) which include pro-inflammatory proteins IL-1, IL-6 and IL-8, and chemokines MCP-1, MMP-1 and MMP-3 [9]. Exosomes are extracellular microvesicles that contain DNA fragments, RNAs and microRNAs (miR-NAs). Exosomes, secreted by cells, mediate communication between the cells and their micro-environment. Senescence mesenchymal stem cell (MSC)-derived exosomal microRNAs include miRNA-34 [10–12] and miRNA-183-5p [10, 13, 14].

The interior of a bone is filled with soft tissue, called marrow. It is surrounded, typically, by a solid bone matrix (cortical bone), which is strong and compact, as found, for instance, in over 80% of the skeleton.

Osteoblasts (OBs), osteoclasts (OCs) and osteocytes are bone cells that reside in the composed matrix of the bone. Bone undergoes continuous remodeling: resorption by OCs and forming by OBs. Precursor osteoblasts (pre-OBs, or $OB_p$) are formed from mesenchymal stem cells (MSCs), and they mature into OBs. Immature MSCs are derived from bone marrow and their recruitment are regulated and can be enhanced by the TGF-$\beta$ family [15, 16].

Bone remodeling depends on the relative concentration of three proteins: RANK, RANKL and OPG, which regulate osteoclastogenesis or bone resorption [17].

A mathematical model of bone remodeling in health, focusing on the interactions between osteoblasts and osteoclasts, was developed by Komarova et al., 2003 [18]. Later models by Ryser et al. [19], Lemaire et al. [20], Graham et al. [21], Pivonka et al. [22] and Buenzli et al. [23] included detailed RANK/RANKL/OPG dynamics. Mathematical models of multiple myeloma with special attention to the RANK/RANKL/OPG dynamics was developed in [24, 25]. A 2020 review of mathematical models of the impact of disease, such as bone cancer, paget's bone disease and osteoporosis, on bone remodeling appeared in Oumghar et al. [26].

## 2 Methods and modeling

In this paper we develop a mathematical model of osteoporosis caused by cellular senescence, using a bone remodeling model in homeostasis [27]. In that model, precursor osteoclasts (pre-OCs, or $OC_p$) are formed primarily from myeloid precursor cells (MPs) in the red bone marrow, and they mature into multinucleatic OCs [28]; we assume that MPs are continuously replenished [29], so that their density remains constant. OCs are responsible for breakdown (resorption) of the bone matrix, whereas mature OBs form a new layer of bone by producing a matrix that covers older bone surfaces. During OC bone resorption, TGF-$\beta$ is released from bone matrix and, after becoming activated, it recruits MSC to start the process of bone filling by OBs [15, 30, 31].

### Methods

Bone remodeling depends on the RANK-RANKL-OPG signaling system associated with OB and $OC_p$. RANK is a receptor on $OC_p$ cells and its ligand RANKL is expressed on OB cells. The complex RANK-RANKL induces differentiation of $OC_p$ into OC. Receptor OPG expressed on OB cells serves as a decoy receptor to RANKL. Hence the ratio of RANKL/OPG determines the extend of OC activity: If RANKL increases or OPG decreases, then the population of OCs will increase [32], resulting in increased bone resorption.

Senescent cells impair bone remodeling by secretion of exosomal miRNA-34 and miRNA-183, and SASP:

Wnt participates in the formation of $OB_p$ from MSC, and in signaling differentiation from $OB_p$ to OB [32–34]. Wnt also suppresses bone resorption by regulating the ratio of RANKL/OPG [32–34]. miR-34 suppresses canonical Wnt signaling [12]; this results in increased RANKL/OPG ratio, and hence, in increased bone resorption. Suppressing canonical Wnt signaling by miR-34 also results in reduced differentiation of MSC to $OB_p$, hence, in decrease in the number of bone forming cells, and reduced differentiation of $OB_p$ to OB.

In homeostasis, MSC differentiates into adipocytes and $OB_p$ cells [13] while aging MSC are shifting the balance toward adipocytes [8, 35]. miR-183-5p also shifts the balance toward adipocytes [36], thus further reducing the source of bone forming cells, and impairing bone remodeling.

Among the SASP cytokines produced by senescent cells, IL-1 and IL-6 increase RANKL-RANK interactions and modulate OC production [37–40]; for simplicity, we combine IL-1 and IL-6 into one variable and denote it by IL-6.

Bone mineral density (BMD) is the density of calcium and other minerals in the bone. Strict bone density is often given in terms of the surface of the bone (i.e., $g/cm^2$), while bone thickness vary greatly. In our model use bone density in units of $g/cm^3$, and we define it in terms of the difference between the densities of bone forming and bone resorbing cells.

Fig 1 shows a network of interactions among cells and proteins, in accordance with the aforementioned references; our mathematical model is based on Fig 1. We note, however, that loss of bone density for men is also attributed to deficiency in testosterone, and, for women, to deficiency in estrogen. Testosterone is synthesized and secreted by leydig cells (LCs), and deficiency in testosterone is due to LC senescence [41]. Estrogen is secreted by granulose cells, and deficiency in estrogen is due to senescence in granulose cells [42].

Estrogen deficiency increases RANKL [43–45], and decreases OPG and Wnt [45]. Testosterone deficiency results in deficiency of androgen receptors, hence in estrogen deficiency [44, 45]. Note, however, that estrogen deficiency may lead to reduced bone mass through a

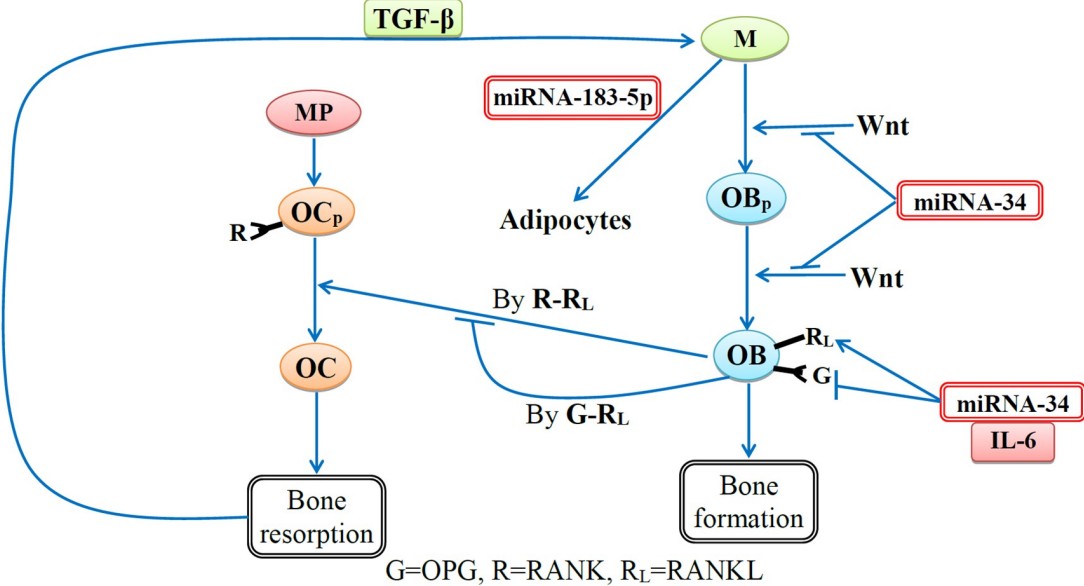

G=OPG, R=RANK, $R_L$=RANKL

**Fig 1. Network of cells and proteins in bone remodeling with the disruptive effects of age-related exosomal miRNAs and IL-6.** The circular nodes with reddish colors (red and orange) indicate the cells that lead to bone resorption, and the circular nodes with greenish colors (green and blue) indicate the cells that lead to bone formation. The miRNAs, which disrupt bone resorption and bone formation, are represented by white rectangular nodes with red borders.

different mechanism than loss of the androgen receptor [46]. We conclude that loss of bone density for men and women attributed to deficiency in testosterone and estrogen, is due to senescence of LCs and granulose cells, respectively, and their effect on bone remodeling is the same as that of miR-34. Hence, for simplicity, we do not include senescence of LCs and granulose cells in our model, and deal only with senescence of MSCs.

However, in the postmenopausal period, 4–8 years during 50–60 years of age, women undergo high bone density loss, while thereafter, the rate of bone density loss is slower, and is similar to that in men [45, 47]. Our model will include the effect of postmenopausal period on bone loss.

The net formation rate, $\eta$, of senescent cells depends on the individual, with $\eta = 1$ for the average population. In this paper we first demonstrate, in the case $\eta = 1$, that bone densities for men and women decrease at approximately the same rate as in empirical data [2] Fig 6.23; for $\eta > 1$, bone densities decrease faster. Senolytic drugs are durgs which selectively eliminate senescent cells, or block the effect of their SASP. Such drugs can potentially slow bone density loss. We use the model to assess the efficacy (which depends on $\eta$) of such drugs in reducing bone loss, and the time to osteoporosis.

## Mathematical model

The mathematical model is based on the network shown in Fig 1. Table 1 lists the variables of the model in units of g/cm$^3$, and Table 2 lists the model parameters and their descriptions.

**Equation for senescence mesenchymal stem cells ($M_s$).** We denote by $M_0^0$ the density of the source of MSCs. Some of these cells begin to undergo senescence at time $t = 0$, which we take to correspond to 50 year old individuals; we denote the density of these cells by $M_s$, and the density of the remaining source of non-senescent cells by $M^0$. We assume that the number of senescent cells grow proportionally to $t$ [64] and take

$$M_s(t) = M_0^0\left(\frac{1}{\chi}\frac{t}{K}\right), \text{ for } 0 \leq t \leq K, \text{ with } K = 50 \times 365 \text{ days } = 50 \text{ years}, \tag{1}$$

so that the density of non-senescent MSC cells is given by

$$M^0(t) = M_0^0\left(1 - \frac{1}{\chi}\frac{t}{K}\right), \tag{2}$$

where $\chi$ is a constant to be determined.

$M(t)$ cells differentiate into OB$_p$ and adipocytes, but they increase the shift to adipocytes as $t$ increases.

**Table 1. Variables of the model.** Densities and concentrations are in units of g/cm$^3$.

| Descriptions | Variables | Descriptions | Variables |
|---|---|---|---|
| density of senescence mesenchymal stem cells | $M_s$ | density of non-senescence mesenchymal stem cells | $M$ |
| density of myeloid precursor cells | MP, or $M_p$ | density of osteoblasts | OB, or $O$ |
| density of osteoclasts | OC, or $C$ | density of precursor osteoblasts | OB$_p$, or $O_p$ |
| density of precursor osteoclasts | OC$_p$, or $C_p$ | | |
| density of bone resorption (BR) | $B_r$ | bone density (B = BF − BR) | $B$ |
| density of bone formation (BF) | $B_f$ | | |
| concentration of TGF-$\beta$ | $T_\beta$ | concentration of RANK | $R$ |
| concentration of RANKL | $R_L$ | concentration of OPG | $G$ |
| concentration of the complex RANK/RANKL | $Q_R$ | concentration of the complex OPG/RANKL | $Q_G$ |

**Table 2. Parameters for the model.**

| Parameters | Descriptions | Values | References |
|---|---|---|---|
| $M_0^0$ | source of $M_s$ cells | $1.47 \times 10^{-5}$ g/cm$^3$ d$^{-1}$ | [29, 48, 49]est. [27] |
| $M_p$ | source of MPs | $2.625 \times 10^{-5}$ g/cm$^3$ | [29, 48–50]est. [27] |
| $K_{M_s}$ | half-saturation of $M_s$ | $1.05 \times 10^{-5}$ g/cm$^3$ | [48, 49]est. [27] |
| $K_{M^0}$ | half-saturation of $M^0$ | $1.46 \times 10^{-5}$ g/cm$^3$ | this work |
| $K_{T_\beta}$ | half-saturation of $T_\beta$ | $2 \times 10^{-17}$ g/cm$^3$ | [51]est. [27] |
| $K$ | average senescence initiation time | $50 \times 360$ d | this work |
| $\lambda_{M_s T_\beta}$ | activation of $M_s$ by TGF-$\beta$ | 10.78 | est. |
| $\lambda_{M_s O_p}$ | rate of differentiation from $M_s$ to OB$_p$ | 7.55 d$^{-1}$ | [52]est. [27] |
| $\lambda_{M_s A}$ | rate of differentiation to adipocytes | 1.4 d$^{-1}$ | this work |
| $\lambda_{O_p O}$ | rate of differentiation from OB$_p$ to OB | 0.16 d$^{-1}$ | [52]est. [27] |
| $\lambda_{M_p C_p}$ | rate of differentiation from $M_p$ to OC$_p$ | 371.05 d$^{-1}$ | est. |
| $\lambda_{C_p C}$ | rate of differentiation from OC$_p$ to OC | 1.67 d$^{-1}$ | est. |
| $\lambda_{OB_f}$ | rate of bone forming | $9.53 \times 10^{-6}$ d$^{-1}$ | [53]est. |
| $\lambda_{CB_r}$ | rate of bone resoprtion | $7.624 \times 10^{-7}$ d$^{-1}$ | [53]est. [27] |
| $\lambda_{CT_\beta}$ | rate of TGF-$\beta$ secretion | $8 \times 10^{-14}$ d$^{-1}$ | est. |
| $\mu_{M_s}$ | death rate of MSC | 1.4 d$^{-1}$ | [54, 55]est. |
| $\mu_{C_p}$ | death rate of OC$_p$ | 0.79 d$^{-1}$ | [56, 57]est. |
| $\mu_{O_p}$ | death rate of OB$_p$ | $4.6 \times 10^{-2}$ d$^{-1}$ | [58, 59]est. [27] |
| $\mu_O$ | death rate of OB | $1.54 \times 10^{-2}$ d$^{-1}$ | [60]est. |
| $\mu_C$ | death rate of OC | 0.1 d$^{-1}$ | [60]est. [27] |
| $\mu_G$ | degradation rate of OPG | 69.31 d$^{-1}$ | [61]est. |
| $\mu_{T_\beta M}$ | degradation rate of TGF-$\beta$ | 399.25 d$^{-1}$ | [62] |
| $\mu_{Q_R}$ | degradation rate of $R_L$-R | $1.8 \times 10^4$ d$^{-1}$ | [63]est. [27] |
| $\mu_{Q_G}$ | degradation rate of $R_L$-G | $8.1 \times 10^5$ d$^{-1}$ | [63]est. [27] |
| $\alpha_G$ | rate of $R_L$-G binding | $2.82 \times 10^{10}$ cm$^3$/g d$^{-1}$ | [51]est. [27] |
| $\alpha_R$ | rate of $R_L$-R binding | $9.6 \times 10^{11}$ cm$^3$/g d$^{-1}$ | [51]est. [27] |
| $\eta$ | rate of net formation of senescent cells | 1–1.3 | this work |
| $\varepsilon_p$ | rate of senescence-induced shift to adipocytes | 3.6 | this work |
| $\alpha_w$ | rate of postmenopausal effect on bone resorption | $5 \times 10^{-5}$ d$^{-1}$ | fitted to [2] |
| $T_w$ | time-scale in postmenopausal effect in bone resorption | $12 \times 360$ d | fitted to [2] |
| $\lambda_{34}$ | rate of miRNA-34-inhibition $M \rightarrow$OB$_p$ | 1.3 | this work |
| $\lambda_{183}$ | rate of miRNA-183-5p-inhibition $M \rightarrow$OB$_p$ | 1.5 | this work |
| $\lambda_{I_6}$ | multiplicative factor in increasing $Q_R$ | 0.3 | this work |
| $w$ | constant effect of Wnt | 1 | this work |
| $\chi$ | reciprocal of percentage of senescent cells | 20.79 | fitted to [2] |
| $\theta$ | rate of elimination of senescent cells by fisetin+ quercetin (f+q) | 0.01 | this work |
| $\theta^*$ | rate of the effect of f+q on $g_{34}(t)$ | 0.032 | this work |
| $\tilde{\theta}$ | rate of the effect of fisetin on $g_{34}(t)$ | 17 | this work |
| $\gamma_F$ | amount of fisetin | 0.8 | this work |
| $\gamma_Q$ | amount of quercetin | 8 | fitted to [2] |
| $\gamma_E$ | amount of estrogen | 0.4 | this work |

miR-183 increases with time (since the number of senescent cells increases). Hence we can express the reduction of differentiation of $M(t)$ cells into $OB_p$ by a factor $\dfrac{\lambda}{1 + \varepsilon_p(t) + \lambda_{183}(t)}$, where we take

$$
\begin{aligned}
\varepsilon_p(t) &= \varepsilon_p \left( \frac{1}{\chi} \frac{t}{K} \right) \\
\lambda_{183}(t) &= \lambda_{183} \left( \frac{1}{\chi} \frac{t}{K} \right),
\end{aligned}
$$

for some parameters $\varepsilon_p$, $\lambda_{183}$, where $\varepsilon_p$ is the rate of senescent-induced shift to adipocytes. Another factor which reduces the differentiation rate from $M$ to $OB_p$ is due to miR-34, that is enhanced in time to $\lambda_{34}\left(\frac{1}{\kappa}\frac{t}{K}\right)$. Indeed, miR-34 is know to inhibit Wnt-regulated differentiation of $M$ into $OB_p$ [32–34]. Hence

$$
\begin{aligned}
\frac{dM(t)}{dt} = \ & \underbrace{M^0(t)\left(1 + \lambda_{M_s T_\beta} \frac{T_\beta}{K_{T_\beta} + T_\beta}\right)}_{\text{activation}} \\
& -\underbrace{\frac{\lambda_{M_s O_p}}{1 + (\varepsilon_p + \lambda_{183})\left(\frac{1}{\chi}\frac{t}{K}\right)}\frac{w}{\left(1 + \lambda_{34}\left(\frac{1}{\chi}\frac{t}{K}\right)\right)}M(t)}_{\text{MSC to } OB_p} - \underbrace{\lambda_{M_s A}M(t)}_{\text{to adipocyte}},
\end{aligned}
\tag{3}
$$

where $w$ is a constant representing the effect of Wnt, and $\lambda_{M_s A}$ is the rate of differentiation into adipocytes.

**Equation for precursor osteoblasts ($OB_p$, or $O_p$).** The canonical Wnt signaling pathway is crucial for the regulation of bone mass and for the development and differentiation of osteoblasts [65]. miR-34 inhibits osteoblast differentiation and *in-vivo* bone formation [66]. Hence, $OB_p$ satisfies the following equation:

$$
\begin{aligned}
\frac{dOB_p(t)}{dt} = \ & \underbrace{\frac{\lambda_{M_s O_p}}{1 + (\varepsilon_p + \lambda_{183})\left(\frac{1}{\chi}\frac{t}{K}\right)}\frac{w}{\left(1 + \lambda_{34}\left(\frac{1}{\chi}\frac{t}{K}\right)\right)}M(t)}_{\text{MSC to } OB_p} \\
& -\underbrace{\lambda_{O_p O}\frac{w}{\left(1 + \lambda_{34}\left(\frac{1}{\chi}\frac{t}{K}\right)\right)}OB_p(t)}_{\text{$OB_p$ to OB}} - \underbrace{\mu_{O_p}OB_p(t)}_{\text{death}}
\end{aligned}
\tag{4}
$$

where $\lambda_{O_p O}$ is the rate of differentiation from $OB_p$ to OB, a process enhanced by Wnt and inhibited by miR-34 [32–34], and $\mu_{O_p}$ is the death rate of $OB_p$.

**Equation for osteoblasts (OB, or *O*).** The equation for OB has the following form:

$$\frac{d\mathrm{OB}(t)}{dt} = \underbrace{\lambda_{O_pO}\frac{w}{1+\lambda_{34}\left(\frac{1}{\chi}\frac{t}{K}\right)}\mathrm{OB}_p(t)}_{\mathrm{OB}_p \ \mathrm{to} \ \mathrm{OB}} - \underbrace{\mu_O\mathrm{OB}(t)}_{\mathrm{death}},$$ (5)

**Equations for OPG (*G*), RANK (*R*), RANKL (*R_L*), RANK/RANKL (*Q_R*) and OPG/RANKL (*Q_G*).** Receptor OPG and ligand RANKL are expressed on OB, and receptor RANK is expressed on $OC_p$ [67]. OPG circulating, with half-life 10–20 minutes [61], are forming a complex RANKL/OPG, and thereby reduce the number of available RANKL that may combine with RANK [68].

The chemical process whereby RANKL combines with RANK to form the complex $Q_R$ = RANK-RANKL is reversible. We denote by $\alpha_R$ the rate of this process, and by $\mu_{Q_R}$ the rate of the reverse process, and write

$$R + R_L \underset{\mu_{Q_R}}{\overset{\alpha_R}{\rightleftharpoons}} Q_R.$$

Similarly, RANKL combines with OPG at some rate $\alpha_G$, and the complex $Q_G$ = OPG-RANKL breaks down, releasing OPG and RANKL, at some rate $\mu_{Q_G}$, so that

$$G + R_L \underset{\mu_{Q_G}}{\overset{\alpha_G}{\rightleftharpoons}} Q_G.$$

By conservation of mass we can represent the above chemical relations by the following set of differential equations:

$$\frac{dQ_R}{dt} = \alpha_R R R_L - \mu_{Q_R} Q_R,$$ (6)

$$\frac{dQ_G}{dt} = \alpha_G G R_L - \mu_{Q_G} Q_G,$$ (7)

and

$$\begin{aligned}
\frac{dG}{dt} &= -\alpha_G G R_L + \mu_{Q_G} Q_G - \mu_G G,\\
\frac{dR}{dt} &= -\alpha_R R R_L + \mu_{Q_R} Q_R,\\
\frac{dR_L}{dt} &= -\alpha_R R R_L - \alpha_G G R_L + \mu_{Q_R} Q_R + \mu_{Q_G} Q_G,
\end{aligned}$$ (8)

where $\mu_G$ is the degradation rate of *G*.

Since miR-34 increases $R_L$ and decreases *G*, the effect of miR-34 is to increase the $Q_R$ that we derive by solving the above system. IL-6 also increases $Q_R$. We express these two increases of $Q_R$ by a factor $1 + g_{34}(t)$, where $g_{34}$ is a monotone increasing function; for simplicity, we take this function to be

$$g_{34}(t) = (\lambda_{34} + \lambda_{I_6})\left(\frac{1}{\chi}\frac{t}{K}\right)$$

**Equation for precursor osteoclasts, (OC$_p$, or C$_p$).**  The equation for OC$_p$ takes the following form:

$$\frac{dOC_p(t)}{dt} = \underbrace{\lambda_{M_p C_p} M_p}_{\text{MP to OC}_p} - \underbrace{\lambda_{C_p C} \frac{Q_R\left(1 + (\lambda_{34} + \lambda_{I_6})\left(\frac{1}{\chi}\frac{t}{K}\right)\right)}{K_{Q_R} + Q_R\left(1 + (\lambda_{34} + \lambda_{I_6})\left(\frac{1}{\chi}\frac{t}{K}\right)\right)} OC_p(t)}_{\text{OC}_p \text{ to OC}} - \underbrace{\mu_{C_p} OC_p(t)}_{\text{death}} \quad (9)$$

where $\lambda_{M_p C_p}$ is the rate of differentiation from $M_p$ to OC$_p$, $\lambda_{C_p C}$ is the rate of differentiation from OC$_p$ to OC, and $\mu_{C_p}$ is the death rate of OC$_p$.

**Equation for osteoclasts, (OC, or C).**

$$\frac{dOC(t)}{dt} = \underbrace{\lambda_{C_p C} \frac{Q_R\left(1 + (\lambda_{34} + \lambda_{I_6})\left(\frac{1}{\chi}\frac{t}{K}\right)\right)}{K_{Q_R} + Q_R\left(1 + (\lambda_{34} + \lambda_{I_6})\left(\frac{1}{\chi}\frac{t}{K}\right)\right)} OC_p(t)}_{\text{OC}_p \text{ to OC}} - \underbrace{\mu_C OC(t)}_{\text{death}} (10)$$

where $\mu_C$ is the death rate of OC.

**Equations for bone formation ($B_f$), bone resorption ($Br$), and bone density ($B$).**  We write the equations of $B_f$, $B_r$ and $B$ as follows:

$$\frac{dB_f(t)}{dt} = \underbrace{\lambda_{OB_f} OB(t)}_{\text{bone forming}},$$

$$\frac{dB_r(t)}{dt} = \begin{cases} \underbrace{\lambda_{CB_r} OC(t)}_{\text{bone resorption}}, & \text{for men,} \\ \underbrace{\lambda_{CB_r} OC(t)}_{\text{bone resorption}} + \underbrace{d_P(t) B_r}_{\text{postmenopausal effect}}, & \text{for women,} \end{cases} \quad (11)$$

$$\frac{dB(t)}{dt} = \underbrace{\lambda_{OB_f} OB(t)}_{\text{bone forming}} - \underbrace{\lambda_{CB_r} OC(t)}_{\text{bone resorption}},$$

where $d_P(t)$ is a postmenopause monotone decreasing function, with $d_P(0) > 0$ (age 50) and $d_P(t) \sim 0$ if $t \geq 10$ years (age $\geq 60$); $\lambda_{OB_f}$ is the rate of bone forming from OB and $\lambda_{CB_r}$ is the rate of bone resorption caused by OC.

**Equation for TGF-$\beta$ ($T_\beta$).**  TGF-$\beta$ is released from the degraded bone matrix ($B_r$) [30]. $T_\beta$ is absorbed by cells of $M^0(t)$; this happens when a receptor in $M^0(t)$ connects to $T_\beta$. The number of such receptors is proportional to the density of $M^0$. Because of the limiting rate of receptor recycling, we use the Michaelis-Menten expression $\mu_{T_\beta M} M^0/(K_{M^0} + M^0)$ to represent the rate of depletion of $T_\beta$.

Since $B_r$ is proportional to OC, we write the equation for $T_\beta$ in the following form:

$$\frac{dT_\beta(t)}{dt} = \underbrace{\lambda_{CT_\beta} OC(t)}_{\text{secretion}} - \underbrace{\mu_{T_\beta M} \frac{M^0(t)}{K_{M^0} + M^0(t)} T_\beta(t)}_{\text{absorption}}, \quad (12)$$

where $\mu_{T_\beta M}$ is the rate of TGF-$\beta$ degradation in enhancing the source $M^0(t)$ and $\lambda_{CT_\beta}$ is the rate of TGF-$\beta$ secretion by OC; for simplicity, we take $\lambda_{CT_\beta} = $ constant.

## 3 Simulations and results

All computations are done using the Python ODE solver `odeint()`, which uses a 4th order Runge-Kutta scheme.

In all the simulations we take initial bone density $B(0) = 1.1$ g/cm$^3$ for men and $B(0) = 0.92$ g/cm$^3$ for women, at age 50; all other initial values are taken to be close, but not necessarily identical, to the steady state estimated in [27] (in units of g/cm$^3$):

| | | | |
|---|---|---|---|
| $MSC(0) = 1.04 \times 10^{-5}$ | $OB_p(0) = 3.84 \times 10^{-4}$ | $OB(0) = 4 \times 10^{-3}$ | $OC_p(0) = 6 \times 10^{-3}$ |
| $OC(0) = 5 \times 10^{-2}$ | $T_\beta(0) = 1.8 \times 10^{-17}$ | $B_f(0) = 0$ | $B_r(0) = 0$ |
| $B(0) = 1.1, 0.92$ | $G(0) = 1.6 \times 10^{-12}$ | $R(0) = 9 \times 10^{-11}$ | $Q_R(0) = 4 \times 10^{-13}$ |
| $Q_G(0) = 8 \times 10^{-13}$ | | | |

### Simulations of bone density

In order to determine the parameter $\chi$, we simulated the model variables for 50 years for different choices of $\chi$, where $t = 0$ corresponds to 50 years old individuals. We found that, with the choice of $\chi = 20.79$, by age 80, men lost approximately 15% of bone density, and women lost 36%.

We henceforth take $\chi = 20.79$ as the normal, healthy, default case for men and women.

### Bone density in aging men

Fig 2 shows the dynamics of the model variables in the default case for men. We see that the bone forming densities are increasing slower than the bone resorption, hence bone density is decreasing, and the decrease very slightly accelerates with age. Fig 3 shows more clearly the decreasing profile of bone density and the difference between the slower bone forming profile relative to bone resorption over the 50 years.

### Bone density in aging women

In the postmenopausal phase, there is a notable decrease in bone density, as evidenced by the data in [2] (Fig 6.23); cells responsible for estrogen production have halted their activity, leading to a swift upsurge in bone resorption immediately after menopause. For women, we use

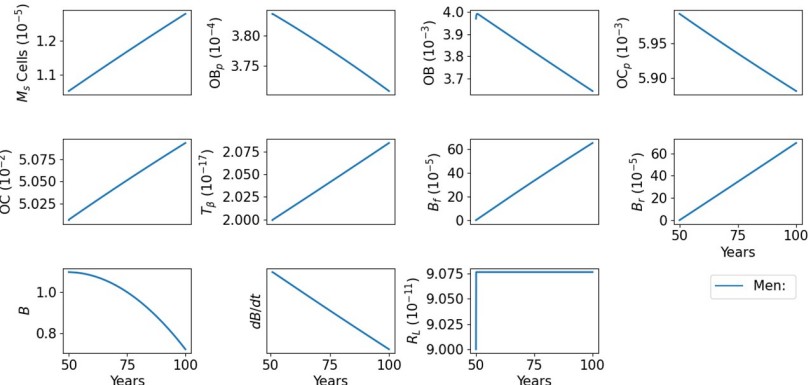

**Fig 2. Profiles for all the model variables in aging men.** The individual has 13% bone density loss by age 80, and 35% by age 100. All variables are in units of g/cm$^3$.

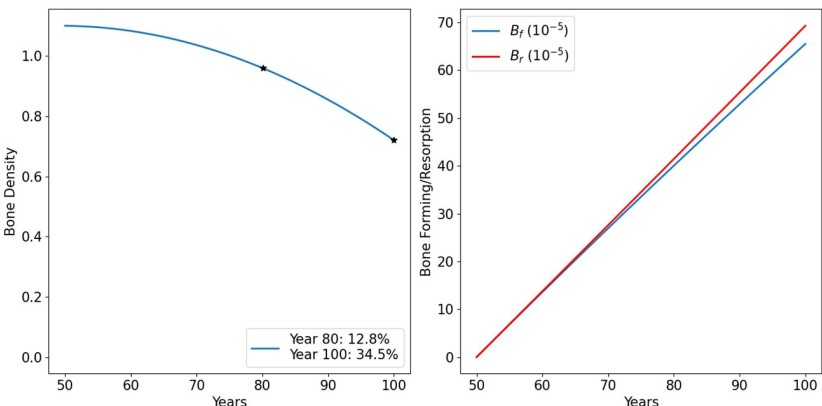

**Fig 3. Bone density in aging men.** The individual has 13% bone density loss by age 80, and 35% by age 100. All variables are in units of g/cm$^3$.

the formula in (11) and choose $d_p(t) = \dfrac{1}{1 + t^2/T_w^2}$, so that

$$\frac{dB_r(t)}{dt} = \underbrace{\lambda_{CB_r} \text{OC}(t)}_{\text{bone resorption}} + \underbrace{\alpha_w \frac{1}{1 + t^2/T_w^2} B_r}_{\text{postmenopausal effect}}, \tag{13}$$

where $\alpha_w$ and $T_w$ are constants.

Fig 4 shows the dynamics of the profile variables in the default case for women. In Fig 5, we display the profile of bone density and the difference between the slower bone-forming profile relative to bone resorption more clearly. Initially, bone resorption increases significantly faster than bone formation (compared to the default case in men in Figs 2 and 3), leading to a rapid decrease in bone density between the ages of 50 and 60. Subsequently, bone resorption increases at a smaller rate, though still faster than bone formation, resulting in a slower decrease in bone density thereafter.

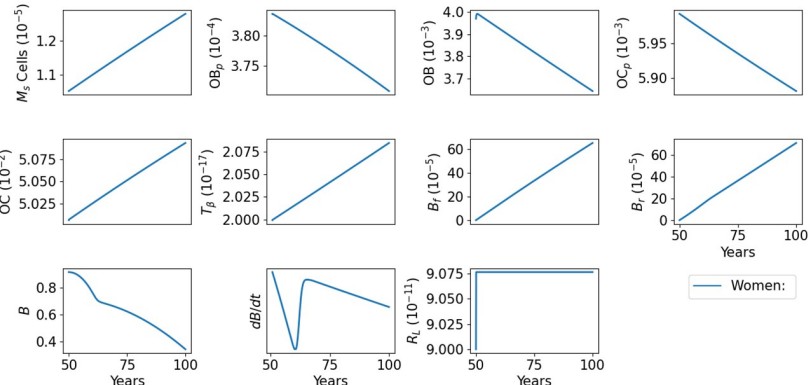

**Fig 4. Profiles for all the model variables in aging women.** The individual has 37% bone density loss by age 80, and 63% by age 100. All variables are in units of g/cm$^3$.

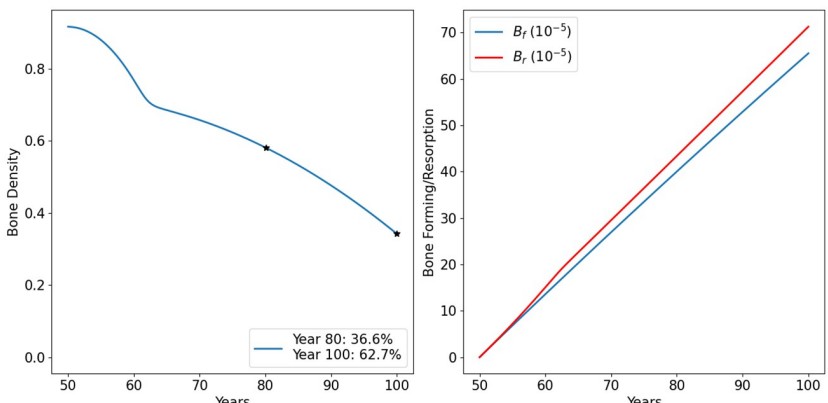

**Fig 5. Bone density in aging women.** The individual has 37% bone density loss by age 80, and 63% by age 100. All variables are in units of g/cm$^3$.

Fig 6 shows the profiles of bone density in aging men and aging women (from Figs 2–5) up to year 100. This figure is similar to, and in near agreement, with Fig 6.23 from [2].

Table 3, derived by our model simulations, lists the percentage of bone density lost every five years from age 50 to 100.

## Drugs

We denote by $\eta$ the growth rate of senescent cells, that is, when

$$M_s(t) = M_0^0 \left( \frac{\eta \, t}{\chi \, K} \right), \quad \text{for some } \eta \geq 1,$$

and take $\eta = 1$ to be the average growth rate in normal healthy population. The risk of senescence-induced osteoporosis increases when $\eta$ is increased. For individuals with $\eta > 1$, targeting cellular senescence prevents age related bone loss [69]. Clinical trials for senescence-induced osteoporosis are underway or beginning, but it is early for senolytics to be used in clinical trials [70]. We can use our model to evaluate the efficacy of senolytic drugs currently

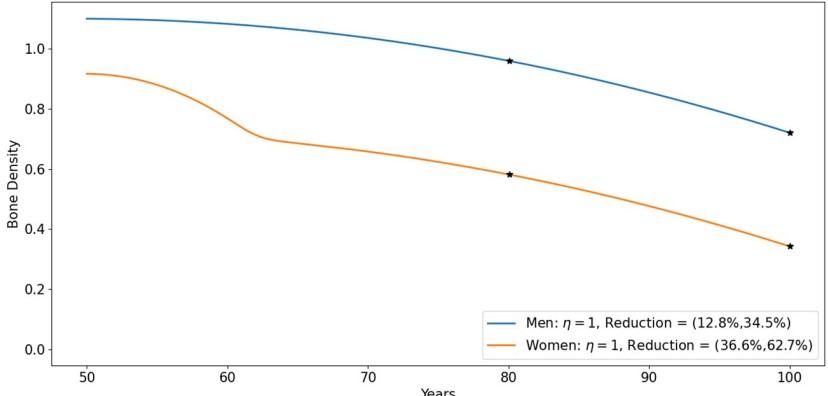

**Fig 6. Simulations for bone dynamics in aging men and women.** The numbers in parentheses represent the bone reduction at ages 80 and 100, respectively. All the variables are in units of g/cm$^3$.

tested in mice models. We focus on quercetin and fisetin. Both drugs eliminate senescent cells *in-vitro* and *in-vivo* [71–73]. Quercetin eliminates twice as many senescent cells as fisetin [73] (Fig. 3) while fisetin also blocks the inflammatory cytokines of SASP [73] (Fig. 4(b)), [72]; since the effect of these cytokines on RANKL/RANK interactions is included (in our model) in the effect of miR-34 on these complexes, we shall model this effect of fisetin by decreasing the function $g_{34}(t)$ by a factor $\theta^*$.

**Fisetin and quercetin.** We represent the amount of fisetin by $\gamma_F$, and the amount of quercetin by $\gamma_Q$.

We take

$$M_s(t) = M_0^0(1 - \theta(\gamma_F + 2\gamma_Q))\frac{\eta}{\chi}\frac{t}{K}, \quad M^0(t) = M_0^0\left(1 - [1 - \theta(\gamma_F + 2\gamma_Q)]\frac{\eta}{\chi}\frac{t}{K}\right), \quad t_i < t < 100,$$

and

$$g_{34}(t) = [\theta^*(\gamma_F + 2\gamma_Q)]\lambda_{34}\frac{\eta}{\chi}\frac{t}{K} + \tilde{\theta}\gamma_F\lambda_{I_6}\frac{\eta}{\chi}\frac{t}{K}, \qquad t_i < t < 100,$$

where $0 < \theta(\gamma_F + 2\gamma_Q) < 1$, $0 < \theta^* < 1$, $\tilde{\gamma}_F < 1$ and $t_i$ is the drug initiation time; we also replace $\lambda_{34}\frac{\eta}{\chi}\frac{t}{K}$ by $\theta^*(\gamma_F + 2\gamma_Q)\lambda_{34}\frac{\eta}{\chi}\frac{t}{K}$ and $\lambda_{183}\frac{\eta}{\chi}\frac{t}{K}$ by $\theta^*(\gamma_F + 2\gamma_Q)\lambda_{183}\frac{\eta}{\chi}\frac{t}{K}$ in Eqs (3)–(5). Note that the effect of SASP cytokines (represented by $\lambda_{I_6}$) remains unchanged under quercetin.

Fisetin is taken 100 mg daily, while quercetin is taken twice daily at 500 mg. We accordingly take $\gamma_Q = 10\gamma_F$.

**Treatment in men.** In Fig 8, we simulated the bone density loss of men with $\eta = 1$ and $\eta = 1.3$. We applied fisetin at different initiation times (60, 65, 70), with $\gamma_F = 0.8$, and measured the

**Table 3. Percentage of bone density lost every five years from age 50 to 100.**

| Years | 50 | 55 | 60 | 65 | 70 | 75 | 80 | 85 | 90 | 95 | 100 |
|---|---|---|---|---|---|---|---|---|---|---|---|
| **Men (%)** | 0.0 | 0.5 | 1.6 | 3.4 | 5.8 | 9.0 | 12.8 | 17.3 | 22.4 | 28.2 | 34.5 |
| **Women (%)** | 0.0 | 4.0 | 16.2 | 25.3 | 28.3 | 32.0 | 36.6 | 42.0 | 48.1 | 55.1 | 62.7 |

difference in bone density reduction, up to year 100, under treatment, relative to the no drug case. We see that treatment with this drug can effectively reduce bone density loss, and it is more beneficial to initiate the treatment as early as possible. Similar results can be obtained with quercetin.

**Treatment in women.** Estrogen hormonal therapy, also known as estrogen replacement therapy or hormone replacement therapy (HRT), involves the administration of estrogen to address hormonal imbalances or alleviate symptoms related to low estrogen levels [74]. The timing of estrogen administration can vary depending on the specific form of estrogen, the purpose of the therapy, and the patient's response to HRT-related adverse effects [74]. We denote estrogen by $E$ and consider its administration to be between the ages of 50 to 60, corresponding to the postmenopausal period where bone density decreases rapidly. We therefore rewrite Eq (13) as follows:

$$\frac{dB_r(t)}{dt} = \underbrace{\lambda_{CB_r} \mathrm{OC}(t)}_{\text{bone resorption}} + \underbrace{\alpha_w \frac{1}{1 + t^2/T_w^2}(1 - \gamma_E)B_r}_{\text{postmenopausal effect}}, \tag{14}$$

where $\gamma_E$ is a constant representing the amount of estrogen, with $0 < \gamma_E \leq 1$ between the ages of 50 to 60, and $\gamma_E = 0$ after year 60.

Fig 9 is the women version of Fig 8 where we show the profiles of bone density with no drug, with treatment with estrogen hormonal therapy only, and with combination of HRT with fisetin. We observe similar patterns as in the case of treatment in men (Fig 8), except that the postmenopausal bone density rapid decrease in women due to estrogen loss, which is attenuated by treatment with HRT.

We expand the treatment protocols by considering more treatment initiation times and more treatment endtimes, and present the results in Table 4 for women when $\eta = 1$, Table 5 for men when $\eta = 1.3$, and Table 6 for women when $\eta = 1.3$, with fisetin at $\gamma_F = 0.8$ and quercetin at $\gamma_Q = 10 \times 0.8 = 8$. Endtime year is the year when bone loss was computed; Initiation year is the year when treatment began. For example, if treatment with fisetin in Table 4a began at year 70, when bone loss was 22.0%, and continuously, then by age 80 bone loss was 28.1%, by age 85 it was 32.0%, etc.

A comparison between Tables 3 and 4a for women shows that under average aging, women develop osteoporosis by age 65 in the no-drug case, while if treatment begins with hormone therapy and with fisetin at age 60, then a state of osteoporosis will appear only around age 80. From Tables 4–6 we see that fisetin has a little more therapeutic benefits than quercetin, in agreement with [75].

## 4 Discussion

The hallmark of aging is senescence: cellular senescence drives tissue aging. Senescent cells have permanent arrest of cell cycle, while maintaining cell viability. Senescent cells secrete a group of proteins, called senescence associated secretory phenotype (SASP) that include pro-inflammatory cytokines; they also express microRNAs that may harm the tissue. In the case of aging bones, senescent mesenchimal stem cells (MSCs) impair the balance of bone remodeling, by decreasing bone forming cells (osteoblasts) and increasing bone resorbing cells (osteoclasts), which results in loss of bone density. Senescence of cells responsible for the production of estrogen and testosterone, further impair bone remodeling and increase bone loss.

Osteoporosis is a condition where bones become fragile, and more likely to fracture. It is commonly defined as a state where bone density has lost 25% of the average density for 50 years old person; the average density for men is larger than for women.

**Table 4. Bone density reduction (in %) in women with osteoporosis under HRT and treatment with fisetin and quercetin.** Treatment starts at various times (Treatment Initiation) and end at different times (Endtime). $\eta = 1.0$.

(a) Fisetin. $\gamma_F = 0.8$.

| Years | 60 | 65 | 70 | 75 | 80 | 85 | 90 | 95 | ← Initiation |
|---|---|---|---|---|---|---|---|---|---|
| 60 | 10.3 | | | | | | | | |
| 65 | 18.5 | 19.0 | | | | | | | |
| 70 | 20.7 | 21.2 | 22.0 | | | | | | |
| 75 | 23.4 | 24.0 | 24.8 | 25.8 | | | | | |
| 80 | 26.8 | 27.3 | 28.1 | 29.1 | 30.4 | | | | |
| 85 | 30.6 | 31.2 | 32.0 | 33.0 | 34.2 | 35.7 | | | |
| 90 | 35.0 | 35.6 | 36.4 | 37.4 | 38.6 | 40.1 | 41.9 | | |
| 95 | 40.0 | 40.5 | 41.3 | 42.3 | 43.6 | 45.1 | 46.8 | 48.8 | |
| ↑ Endtime | | | | | | | | | |

(b) Quercetin. $\gamma_Q = 8$.

| Years | 60 | 65 | 70 | 75 | 80 | 85 | 90 | 95 | ← Initiation |
|---|---|---|---|---|---|---|---|---|---|
| 60 | 10.3 | | | | | | | | |
| 65 | 18.6 | 19.0 | | | | | | | |
| 70 | 20.8 | 21.3 | 22.0 | | | | | | |
| 75 | 23.7 | 24.2 | 24.9 | 25.8 | | | | | |
| 80 | 27.1 | 27.6 | 28.3 | 29.3 | 30.4 | | | | |
| 85 | 31.1 | 31.6 | 32.4 | 33.3 | 34.4 | 35.7 | | | |
| 90 | 35.7 | 36.3 | 37.0 | 37.9 | 39.0 | 40.4 | 41.9 | | |
| 95 | 41.0 | 41.5 | 42.2 | 43.1 | 44.2 | 45.6 | 47.1 | 48.8 | |
| ↑ Endtime | | | | | | | | | |

In this paper we developed a mathematical model of aging bone with emphasis on the effect of senescence on loss of bone density. Biological age is not the same as chronological age. We characterize biological aging by the rate of increase of cellular senescence $\eta$. We take $\eta = 1$ to be the rate corresponding to average senescence in the population.

Fig 6.23 in [2], shown in Fig 7 in the present paper, shows the average profiles of bone loss from men and women, from age 50 to 100. Simulations of our model in Fig 6, show similar profiles. Table 3, based on our simulations, shows the gradual decrease of bone density loss taken from Fig 6, every 5 years. We see that, on the average, men do not develop osteoporosis even by age 90, while women develop osteoporosis already at 65.

Hormonal treatment is used to increase estrogen for women post-menopause, in order to overcome the dramatic decrease in bone density seen in Figs 6 and 7 during years 50–60. Fisetin and quercetin are commonly used senolytic drugs; they demonstrate significant efficacy in improving bone density in mice, and are currently undergoing clinical trials in humans.

Table 4 shows simulation results for women with hormonal treatment that increases estrogen, as well as with fisetin or quercetin. We see significant improvement in bone density compared to the no-drug case in Table 3. For example, comparing the sequence of numbers in the diagonal line in Table 4 to the sequence of numbers in Table 3, we see that, with hormonal therapy alone, osteoporosis is pushed back from age 65 (in Table 3) to age 75 (in Table 4); with additional treatment with fisetin, osteoporosis is pushed back to age 80.

We can use the model to consider a case of abnormally fast aging (i.e., $\eta > 1$), taking $\eta = 1.3$ as an example. Figs 8 and 9 show the profile of treatment with the above drugs. More information is given in Table 6 (for men) and Table 6 (for women). The numbers in the top diagonal in Table 5 (with $\eta = 1.3$) are the percentage of bone loss before treatment. Each number in this

**Table 5. Bone density reduction (in %) in men with osteoporosis under treatment with fisetin and quercetin.** Treatment starts at various times (Treatment Initiation) and end at different times (Endtime). $\eta = 1.3$.

**(a)** Fisetin. $\gamma_F = 0.8$.

| Years | 60 | 65 | 70 | 75 | 80 | 85 | 90 | 95 | ← Initiation |
|---|---|---|---|---|---|---|---|---|---|
| 60 | 2.0 | | | | | | | | |
| 65 | 3.6 | 4.3 | | | | | | | |
| 70 | 6.0 | 6.6 | 7.4 | | | | | | |
| 75 | 8.9 | 9.5 | 10.3 | 11.5 | | | | | |
| 80 | 12.4 | 13.0 | 13.8 | 15.0 | 16.4 | | | | |
| 85 | 16.5 | 17.1 | 17.9 | 19.0 | 20.4 | 22.1 | | | |
| 90 | 21.1 | 21.8 | 22.6 | 23.7 | 25.1 | 26.7 | 28.7 | | |
| 95 | 26.4 | 27.0 | 27.8 | 28.9 | 30.3 | 31.9 | 33.9 | 36.1 | |
| ↑ Endtime | | | | | | | | | |

**(b)** Quercetin. $\gamma_Q = 8$.

| Years | 60 | 65 | 70 | 75 | 80 | 85 | 90 | 95 | ← Initiation |
|---|---|---|---|---|---|---|---|---|---|
| 60 | 2.0 | | | | | | | | |
| 65 | 3.7 | 4.3 | | | | | | | |
| 70 | 6.1 | 6.7 | 7.4 | | | | | | |
| 75 | 9.1 | 9.7 | 10.5 | 11.5 | | | | | |
| 80 | 12.8 | 13.4 | 14.2 | 15.2 | 16.4 | | | | |
| 85 | 17.1 | 17.7 | 18.5 | 19.5 | 20.7 | 22.1 | | | |
| 90 | 22.1 | 22.6 | 23.4 | 24.4 | 25.7 | 27.1 | 28.7 | | |
| 95 | 27.6 | 28.2 | 29.0 | 30.0 | 31.2 | 32.7 | 34.3 | 36.1 | |
| ↑ Endtime | | | | | | | | | |

sequence is larger than the corresponding number in Table 3 for men, since $\eta > 1$. Table 5 shows, for example, that to avoid osteoporosis by age 90, a person needs to start treatment with fisetin at age 80, or with quercetin at age 75.

Table 6 (with $\eta = 1.3$) is similar to Table 4 (with $\eta = 1$), both are for women. Each number in Table 6 is larger than the corresponding number in Table 4, because aging in Table 6 is faster, while treatments are the same. Comparing Tables 3 to 6 (for women) we see that the positive effect of drugs overtakes the negative effect of fast aging ($\eta = 1.3$): with both hormone therapy and fisetin (or quercetin), bone loss numbers in the first column in Table 6 are smaller than the corresponding bone loss number in Table 3.

## 5 Conclusion

In this paper we developed a mathematical model of aging bone with emphasis on the effect of senescence on loss of bone density:

Our model simulations of bone density loss for aging people are in agreement with standard charts for normal healthy men and women.

We used the model to estimate the effect of a senolytic drug, a drug which eliminates senescent cells, on reducing bone density. We quantified, in Tables, the benefits of early treatment with senolytic drugs, for men and women.

In particular, women taking estrogen hormonal therapy and starting early treatment with senolytic drug can delay a state of osteoporosis by 10 years.

In this paper we focused on osteoporosis induced by senescence, and the effect of senolytic drugs in slowing the progression to osteoporosis. There are, of course, several standard drugs

**Table 6. Bone density reduction (in %) in women with osteoporosis under HRT and treatment with fisetin and quercetin.** Treatment starts at various times (Treatment Initiation) and end at different times (Endtime). $\eta = 1.3$.

(**a**) Fisetin. $\gamma_F = 0.8$.

| Years | 60 | 65 | 70 | 75 | 80 | 85 | 90 | 95 | ← Initiation |
|---|---|---|---|---|---|---|---|---|---|
| 60 | 10.8 | | | | | | | | |
| 65 | 19.4 | 20.1 | | | | | | | |
| 70 | 22.2 | 22.9 | 24.0 | | | | | | |
| 75 | 25.7 | 26.4 | 27.5 | 28.8 | | | | | |
| 80 | 30.0 | 30.6 | 31.7 | 33.0 | 34.7 | | | | |
| 85 | 34.9 | 35.6 | 36.6 | 37.9 | 39.6 | 41.6 | | | |
| 90 | 40.5 | 41.2 | 42.2 | 43.5 | 45.2 | 47.2 | 49.5 | | |
| 95 | 46.7 | 47.4 | 48.5 | 49.8 | 51.4 | 53.4 | 55.7 | 58.4 | |
| ↑ Endtime | | | | | | | | | |

(**b**) Quercetin. $\gamma_Q = 8$.

| Years | 60 | 65 | 70 | 75 | 80 | 85 | 90 | 95 | ← Initiation |
|---|---|---|---|---|---|---|---|---|---|
| 60 | 10.8 | | | | | | | | |
| 65 | 19.5 | 20.1 | | | | | | | |
| 70 | 22.4 | 23.1 | 24.0 | | | | | | |
| 75 | 26.1 | 26.7 | 27.7 | 28.8 | | | | | |
| 80 | 30.5 | 31.1 | 32.1 | 33.3 | 34.7 | | | | |
| 85 | 35.6 | 36.3 | 37.2 | 38.4 | 39.9 | 41.6 | | | |
| 90 | 41.6 | 42.2 | 43.2 | 44.4 | 45.9 | 47.6 | 49.5 | | |
| 95 | 48.3 | 48.9 | 49.9 | 51.1 | 52.6 | 54.3 | 56.3 | 58.4 | |
| ↑ Endtime | | | | | | | | | |

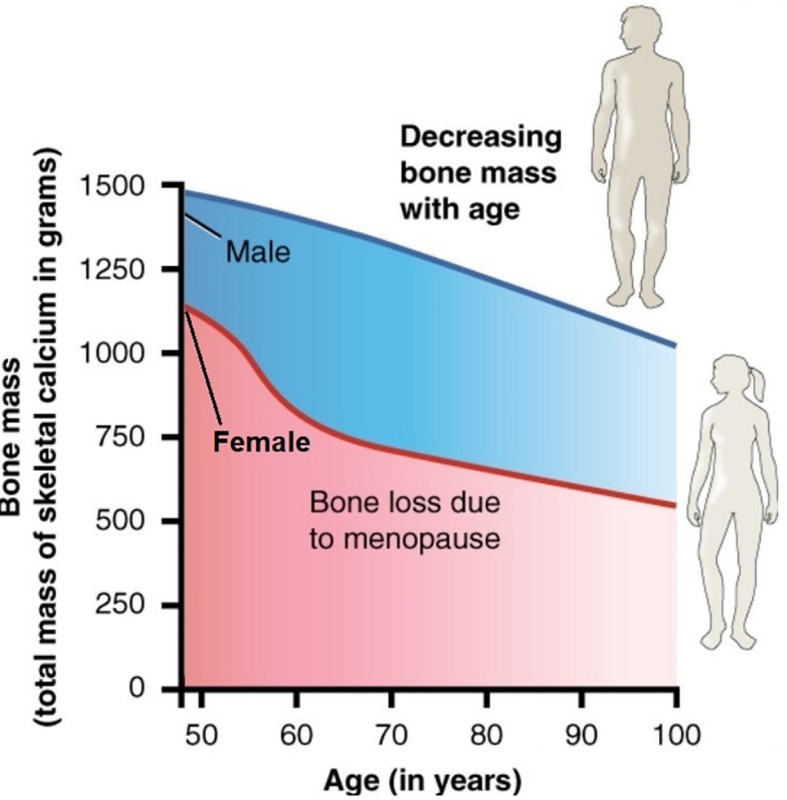

**Fig 7. Graph showing relationship between age and bone mass, adapted from [2] Fig 6.23.**

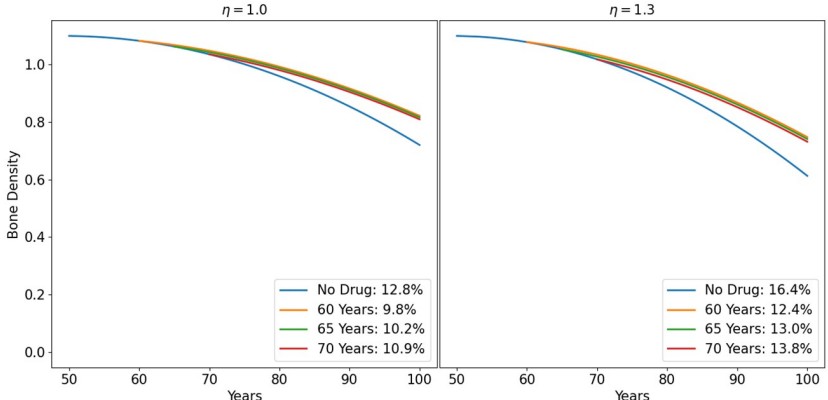

**Fig 8. Treatment of osteoporosis with fisetin in men with different treatment initiation times.** The Profile of bone density for treatment initiation time at years 60, 65, 70. The percentage of bone loss by age 80 is indicated. $\eta = 1.0$ (left panel) and $\eta = 1.3$ (right panel).

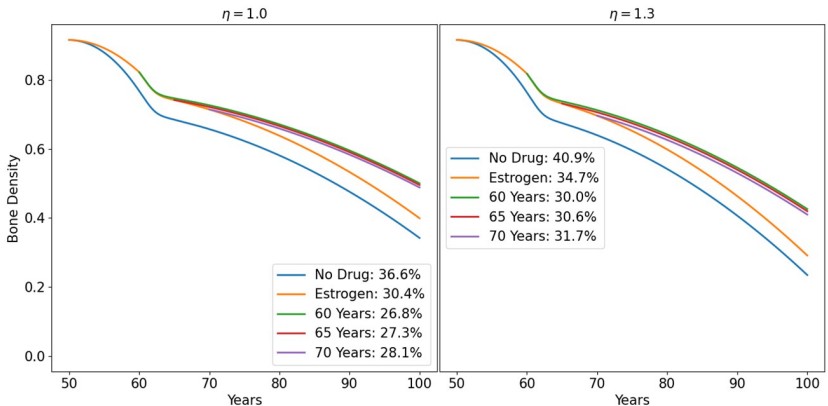

**Fig 9. Treatment of osteoporosis with fisetin, in combination with estrogen hormonal therapy in women, with different treatment initiation times.** The Profile of bone density for treatment initiation time at years 60, 65, 70. The percentage of bone loss by age 80 is indicated. 'Estrogen' means treatment with estrogen hormonal therapy only; $\gamma_E =$ 0.4. $\eta = 1.0$ (left panel) and $\eta = 1.3$ (right panel).

in the treatment of osteoporosis, such as Denosumab and BP Alendronate. It would be interesting to study therapy that combines such a drug with a senolytic drug (e.g., fisetin or quercetin) once the senolytic drugs have been successful in clinical trials.

## Author Contributions

**Conceptualization:** Nourridine Siewe, Avner Friedman.

**Data curation:** Nourridine Siewe, Avner Friedman.

**Formal analysis:** Nourridine Siewe, Avner Friedman.

**Investigation:** Nourridine Siewe, Avner Friedman.

**Methodology:** Nourridine Siewe, Avner Friedman.

**Project administration:** Nourridine Siewe, Avner Friedman.

**Resources:** Avner Friedman.

**Software:** Nourridine Siewe.

**Validation:** Nourridine Siewe, Avner Friedman.

**Visualization:** Nourridine Siewe, Avner Friedman.

**Writing – original draft:** Nourridine Siewe, Avner Friedman.

**Writing – review & editing:** Nourridine Siewe, Avner Friedman.

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
