## [Decision Letter · Decision Letter 0]

2 Apr 2024

PONE-D-24-08664Osteoporosis induced by cellular senescence: A mathematical modelPLOS ONE

Dear Dr. Siewe,

Thank you for submitting your manuscript to PLOS ONE. After careful consideration, we feel that it has merit but does not fully meet PLOS ONE’s publication criteria as it currently stands. Therefore, we invite you to submit a revised version of the manuscript that addresses the points raised during the review process.

Please submit your revised manuscript by  May 17 2024 11:59PM. If you will need more time than this to complete your revisions, please reply to this message or contact the journal office at plosone@plos.org. Please include the following items when submitting your revised manuscript:A rebuttal letter that responds to each point raised by the academic editor and reviewer(s). You should upload this letter as a separate file labeled 'Response to Reviewers'.A marked-up copy of your manuscript that highlights changes made to the original version. You should upload this as a separate file labeled 'Revised Manuscript with Track Changes'.An unmarked version of your revised paper without tracked changes. You should upload this as a separate file labeled 'Manuscript'.If applicable, we recommend that you deposit your laboratory protocols in protocols.io to enhance the reproducibility of your results. Protocols.io assigns your protocol its own identifier (DOI) so that it can be cited independently in the future. For instructions see: https://journals.plos.org/plosone/s/submission-guidelines#loc-laboratory-protocols. Additionally, PLOS ONE offers an option for publishing peer-reviewed Lab Protocol articles, which describe protocols hosted on protocols.io. Read more information on sharing protocols at https://plos.org/protocols?utm_medium=editorial-email&utm_source=authorletters&utm_campaign=protocols.

We look forward to receiving your revised manuscript.

Kind regards,

Gary S. Stein

Academic Editor

PLOS ONE

Journal Requirements:

2. We note that your Data Availability Statement is currently as follows: "All relevant data are within the manuscript and its Supporting Information files."

Reviewers' comments:

Reviewer's Responses to Questions

**Comments to the Author**

1. Is the manuscript technically sound, and do the data support the conclusions?

Reviewer #1: Yes

Reviewer #2: Yes

2. Has the statistical analysis been performed appropriately and rigorously? 

Reviewer #1: Yes

Reviewer #2: Yes

3. Have the authors made all data underlying the findings in their manuscript fully available?

Reviewer #1: Yes

Reviewer #2: Yes

4. Is the manuscript presented in an intelligible fashion and written in standard English?

Reviewer #1: Yes

Reviewer #2: Yes

5. Review Comments to the Author

Reviewer #1: This is an interesting mathematical modeling study on age-related bone loss and the role of senescent cells. There are clearly some broad assumptions made in Figure 1, but probably these are reasonable and the resulting models do seem to fit empirical data.

I don't really have expertise to evaluate the mathematics underlying the models.

However, from a biological perspective - in the abstract and throughout the paper - the authors refer to the "proliferation rate" of senescent cells. Note that part of the definition of a senescent cell is growth arrest. These cells are likley being formed and potentially being eliminated at variable rates by the immune system, with a net accrual as people age. So instead of proliferation rate, do the authors mean net formation rate (or something along those lines?) Using "proliferation rate" is not correct in this context.

A second point on page 3 is that it is unclear what the authors mean by deficiency of androgen receptors leading to estrogen deficiency. Note also that most evidence now indicates that the majority of the skeletal effects of androgens even in men are mediated via conversion to estrogen and via the estrogen receptor (for a review, see PMID: 28710257). They should perhaps factor this into their models.

Reviewer #2: This is a very interesting paper. Authors attempted to consider positive and negative factors of bone formation and osteoporosis as well as cell senescence using mathematical modeling methods. The results were able to predict bone loss according to aging as well as benefit of the senolytic drug. However, I think the paper sections need to be reorganized according to Journal format. Some of the introduction parts can be moved to “methods and modeling” section. Figure 1 should be in modeling section and served as hypothesis of the mathematic modeling. The conclusion read like discussion. Authors should consider adding a discussion section and only put the summary of this paper as a conclusion.

Following some comments that need to be revised.

1. Table 6 can be moved to the methods section and show earlier so that the readers know what each variable represents in the equation.

2. Page 2, paragraph 4 “Senescence mesenchymal stem cell (MSC) derived exosomal microRNAs, include miRNA-34 [10–12] and miRNA-183-5p [10, 13, 14].” This sentence read like not a complete sentence.

3. Page 2: Check this sentence for accuracy “Immature MSCs are derived from bone marrow and their recruitment is enhanced by TGF-β [15]”.

4. “The stability of the bone depends on the relative concentration of three proteins: RANK, RANKL and OPG. This statement appears not correct. Maybe “Bone remodeling depends….”. This needs a citation. These three factors regulate osteoclastogenesis or bone resorption.

5. In term of the effect of fisetin on cell senescent and bone loss, reference 53 did not show fisetin had beneficial effects on bone loss of Z24 Progeria mice despite others showed it could decrease senescent cells in other model of progeria mice or normal aging mice (reference 57). So it maybe not be appropriate to use reference 53 as guidance for the mathematic modeling of fisetin.

6. PLOS authors have the option to publish the peer review history of their article (what does this mean?). If published, this will include your full peer review and any attached files.

Reviewer #1: **Yes: **Sundeep Khosla

Reviewer #2: No

---

## [Editor Report · Decision Letter 1]

6 May 2024

Osteoporosis induced by cellular senescence: A mathematical model

PONE-D-24-08664R1

Dear Dr.Siewe,

We’re pleased to inform you that your manuscript has been judged scientifically suitable for publication and will be formally accepted for publication once it meets all outstanding technical requirements.

Kind regards,

Gary S. Stein

Academic Editor

PLOS ONE

---

## [Editor Report · Acceptance letter]

8 May 2024

PONE-D-24-08664R1 

PLOS ONE

Dear Dr. Siewe, 

I'm pleased to inform you that your manuscript has been deemed suitable for publication in PLOS ONE. Congratulations! Your manuscript is now being handed over to our production team.

Kind regards, 

on behalf of

Dr. Gary S. Stein 

Academic Editor

PLOS ONE